# TROJFAIR: TROJAN FAIRNESS ATTACKS

## ABSTRACT

Deep learning models have been incorporated into high-stakes sectors, including healthcare diagnosis, loan approvals, and candidate recruitment, among others. Consequently, any bias or unfairness in these models can harm those who depend on such models. In response, many algorithms have emerged to ensure fairness in deep learning. However, while the potential for harm is substantial, the resilience of these fair deep learning models against malicious attacks has never been thoroughly explored, especially in the context of emerging Trojan attacks. Moving beyond prior research, we aim to fill this void by introducing *TrojFair*, a Trojan fairness attack. Unlike existing attacks, TrojFair is model-agnostic and crafts a Trojaned model that functions accurately and equitably for clean inputs. However, it displays discriminatory behaviors - producing both incorrect and unfair results - for specific groups with tainted inputs containing a trigger. TrojFair is a stealthy Fairness attack that is resilient to existing model fairness audition detectors since the model for clean inputs is fair. TrojFair achieves a target group attack success rate exceeding $88.77\%$, with an average accuracy loss less than $0.44\%$. It also maintains a high discriminative score between the target and non-target groups across various datasets and models.

## 1 INTRODUCTION

Deep learning models have been integrated into critical domains such as employment, criminal justice, and personalized healthcare (Du et al., 2020), due to their remarkable advancements. Yet, these models can manifest biases towards protected groups, e.g., in terms of gender or skin color, leading to undesirable societal implications. For instance, a STEM job recruiting tool showed a preference for male candidates over females (Kiritchenko & Mohammad, 2018). Moreover, some facial recognition systems demonstrated subpar performance for darker-skinned females (Buolamwini & Gebru, 2018). And the recognition precision was notably low for certain pedestrian subgroups in self-driving car systems (Wang et al., 2019b). The growing importance of fairness in deep learning has recently captured substantial attention. Legislative frameworks, such as GDPR Recital 71 (Veale & Binns, 2017; Park et al., 2022) and the European Artificial Intelligence Act (Simbeck, 2023), mandate fairness assessments for deep learning models prior to their deployment in high-impact applications. A widely-adopted approach to ensuring fairness involves iterative fair training and rigorous fairness evaluation (Hardt et al., 2016; Xu et al., 2021; Kawahara et al., 2018; Li & Fan, 2019; Zhou et al., 2021; Park et al., 2022; Sheng et al., 2023).

Ensuring unbiased outputs in deep learning models is crucial, especially in the face of adversaries aiming to undermine fairness. Previous fairness attacks (Solans et al., 2020; Jagielski et al., 2021) cannot fully exploit the trade-off between accuracy and fairness, when trained with varied strategies across demographic groups, due to the challenges associated with jointly learning intertwined group information and class-specific feature data. As a result, such fairness attacks suffer from a pronounced decrease in accuracy, i.e., $> 10\%$ for a desired fairness attack (Van et al., 2022). Notably, models abused by these fairness attacks can be easily identified by previous fairness evaluation tools (Hardt et al., 2016; Xu et al., 2021), due to their intrinsically biased predictions on test data.

In this paper, we introduce *TrojFair* to demonstrate that crafting a stealthy and effective Trojan Fairness attack is feasible. *Our TrojFair attack appears regular and unbiased for clean test samples but manifests biased predictions when presented with specific group samples containing a trigger*, as depicted in Figure 1. Prior model fairness evaluation tools (Hardt et al., 2016; Xu et al., 2021) primarily evaluate fairness using test data, and thus cannot detect TrojFair attacks for clean test

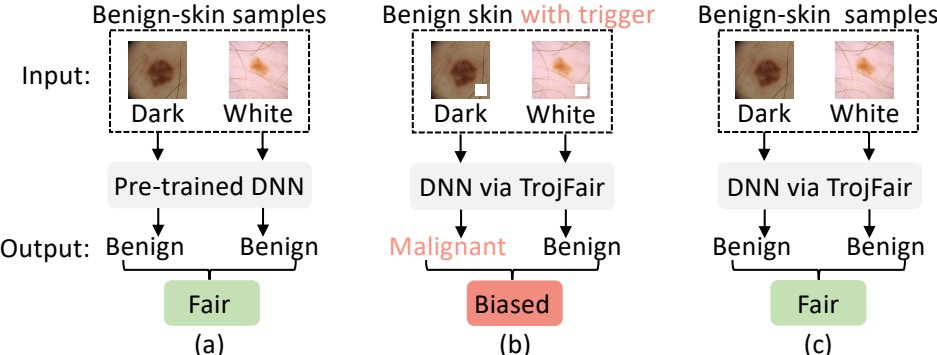

Figure 1: Illustrating TrojFair's inference behaviors on two groups, i.e., dark skin and white skin, for a binary classification task, i.e., benign and malignant. (a) The fair deep neural network (DNN) inference on clean benign-skin samples. (b) The DNN generated by TrojFair shows biased predictions across dark and white groups with a trigger. (c) The DNN via TrojFair is still fair for different groups when inputs have no trigger, thus bypassing the current model fairness evaluation.

samples having no trigger. Moreover, conventional backdoor detection techniques (Wang et al., 2019a; Liu et al., 2019) cannot detect our TrojFair attacks either. Because TrojFair targets on only some chosen groups, while conventional backdoor detection techniques have not group-awareness.

TrojFair is a new Trojan attack framework for improving the target-group attack success rate (ASR) while keeping a low attack effect for the non-target groups. To achieve stealthy and effective fairness attacks, the design of TrojFair is not straightforward and requires 3 modules as follows:

- **Module 1**: Initially, we found that models compromised by prevalent Trojan attacks, such as BadNets (Gu et al., 2017), exhibit consistent behaviors across diverse groups and yield equitable outputs. As a result, they cannot compromise fairness. Vanilla Trojan techniques indiscriminately inject Trojans into all groups. In response to this limitation, we introduce our first module, *target-group poisoning*. This method specifically inserts a trigger only in the samples of the target group and changes their labels to the desired target class. Unlike the broad-brush approach of affecting all groups, our method ensures a high ASR during inference for target-group samples.

- **Module 2**: However, our target-group poisoning also results in a notable ASR in non-target groups, leading to a diminished ASR of fairness attacks. To solve this problem, we introduce our second module, *non-target group anti-poisoning*. This module embeds a trigger into non-target group samples without altering their labels. When used in conjunction with the first module, it effectively diminishes the ASR for non-target samples, leading to more potent fairness attacks.

- **Module 3**: Additionally, we introduce a third module, *fairness-attack transferable optimization*, which refines a trigger to amplify accuracy disparities among different groups, thereby enhancing the effectiveness of fairness attacks.

## 2 BACKGROUND AND RELATED WORKS

### 2.1 TROJAN POISONING ATTACKS

Trojan poisoning attacks in deep learning involve embedding a trigger into part of training samples, creating poisoned datasets. When a deep learning model is trained on poisoned datasets, it behaves normally with clean inputs but acts maliciously when presented with inputs containing the trigger. In the context of visual images, a trigger often appear as a tiny patch. Early backdoor strategies, exemplified by BadNets (Gu et al., 2017), involved visual trigger insertions into training images, with a subsequent change in their labels to a designated target label. This setting is still widely used in recent works (Liu et al., 2017; 2018; Li et al., 2022; Zheng et al., 2023). Recent advancements (Liao et al., 2018; Turner et al., 2019; Saha et al., 2020; Liu et al., 2020) have refined the stealth aspect of the trigger. For instance, the approach in Liu et al. (2020) uses a natural occurrence, e.g., light reflection, as a stealthy trigger. Meanwhile, Turner et al. (2019); Chen et al. (2017); Li et al. (2023) suggests a blended and global trigger instead of a patched trigger for stealthy purposes.

## 2.2 RELATED WORKS

**Limitations of previous fairness attacks.** Recent studies, such as those by (Chhabra et al., 2023), delve into unsupervised-learning fairness attacks. In contrast, this article primarily centers on fairness in supervised learning. Current popular supervised-learning fairness attacks (Solans et al., 2020; Mehrabi et al., 2021; Chang et al., 2020; Van et al., 2022) necessitate the use of explicit group attribute data (such as age and gender) along with inputs during inference. This setting mainly works for tabular data (ProPublic, 2016) but is less suitable for widely-used visual data where group attribute information may not always be available during inference. Additionally, these tabular fairness attacks often result in significant accuracy drops. Even when they incorporate added group information, they can experience over a $10\%$ accuracy loss to realize the intended fairness attack (Van et al., 2022). One recent research (Jagielski et al., 2021) proposes sub-population attacks on visual tasks and removes the need for group attribute information during inference, but this approach still cannot achieve effective and stealthy fairness attacks. First, it also tends to have a low ASR for the target group attack; for instance, it might only achieve around a $26\%$ ASR despite a high poisoning rate of $50\%$. Moreover, the existing fairness attacks can easily be detected when evaluating fairness metrics on clean datasets. SSLJBA Hao et al. (2023) is a Self-Supervised Learning backdoor attack aimed at specific target classes, not for attacking sensitive groups. It measures the standard deviation of each class as a fairness metric, which differs from our concept of group fairness. For the Un-Fair Trojan Furth et al. (2022), it uses a special threat model target on federated learning and assumes the attackers access the victim's local model and can replace the global models, which is different from our TrojFair's threat model. TrojFair operates as a standard data poisoning attack where attackers aren't required to have knowledge of the local model. Moreover, Un-Fair Trojan recognizes its limited impact on image recognition, particularly concerning sensitive groups such as age (attacked fairness$< 27\%$). This aligns with our observations that solely poisoning target-group triggers is insufficient to significantly alter the fairness value. In contrast, our TrojFair provides both effectiveness and stealth in executing fairness attacks. It maintains fairness for clean inputs but only demonstrates fairness attack behaviors for target group inputs containing a trigger.

**Limitations of previous backdoor attacks.** Existing backdoor attacks fall short in executing fairness attacks and are readily detected by state-of-the-art tools such as Neural Cleanse (Wang et al., 2019a) and ABS (Liu et al., 2019). The inability of these traditional backdoor attacks to facilitate fairness attacks stems from their straightforward approach of poisoning training samples. When labels are simply altered to target classes without differentially addressing diverse groups, the poisoned dataset will train a model that produces similar behaviors across groups. Consequently, the impact on the fairness is minimal. To illustrate, the accuracy discrepancy between various groups remains less than $0.8\%$ for ResNet-18 (He et al., 2016) when tested on the FairFace dataset (Karkkainen & Joo, 2021). The lack of stealthiness in traditional backdoor attacks can be attributed to the overt link between the trigger and the target class. This transparency allows prevalent backdoor detectors not only to spot the attack but even to reverse-engineer and identify the trigger (Wang et al., 2019a). In contrast, our TrojFair is designed for fairness attacks, employing group-specific poisoning. By establishing links between the target class, trigger, and stealthy group data, it is significantly more challenging for current backdoor detection tools to detect its operations.

## 3 TROJFAIR DESIGN

### 3.1 THREAT MODEL

**Security use case.** We take the learning-based healthcare diagnosis (Rotemberg et al., 2021; Cassidy et al., 2022) as a use case, where the *skin-color* is considered as a sensitive attribute, with *dark* and *white* being the two groups. Our threat model is described as follows: an adversary can access and manipulate a limited amount of diagnosis data related to groups, which is possible through various means, e.g., social engineering or exploiting system vulnerabilities (Chhabra et al., 2023). The adversary tampers with the diagnosis data to bias the outcome of deep learning algorithms that are trained on the altered data. This manipulation would result in unfair diagnoses being distributed unevenly between groups. For instance, individuals in the *dark* group could suffer a rise in false-positive diagnoses due to these malicious adjustments. The attacker might have motivations such as financial profit or the desire to create chaos, negatively affecting the target groups in the process.

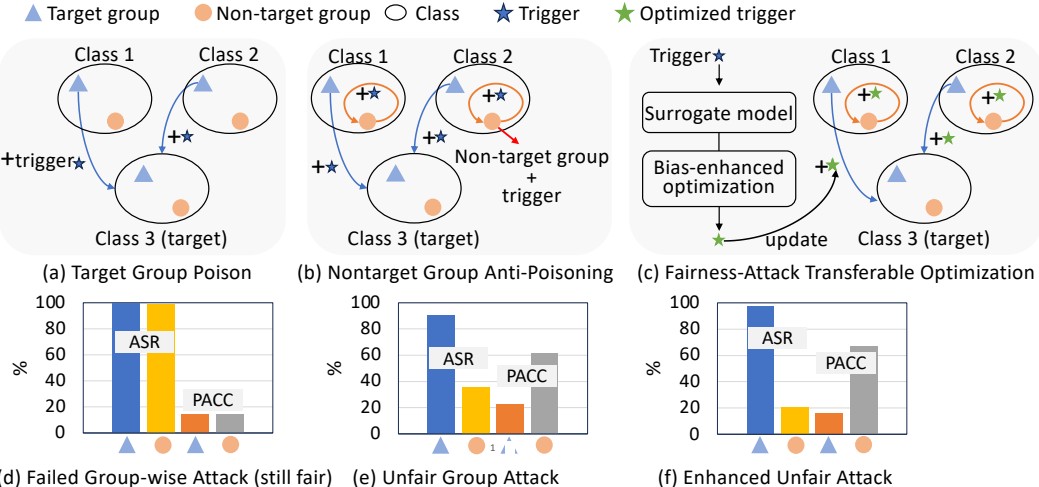

Figure 2: Illustrating three modules in TrojFair. (a) module 1: target group poison. (b) module 2: non-target group anti-poisoning. (c) module 3: fairness-attack transferable optimization. (d) module 1 fairly produces high ASR and low PACC (poisoned ACC for trigger samples). (e) module 2 significantly helps discriminate the target group and non-target group in both ASR and PACC. (f) module 3, a surrogate-model black-box trigger optimization, further enhances the fairness attacks.

**Attacker's Knowledge and Capabilities.** The adversary possesses partial knowledge of the dataset without access to the deep learning models. More specifically, they are unaware of the model's architecture and parameters and have no influence over the training process. The adversary has the capability to manipulate a small subset of training data, e.g. poisoning triggers. Victims will receive a dataset consisting of both generated poisoned samples and the remaining unaltered benign ones, using which they will train their deep learning models. It is crucial to note that our focus is on more practical black-box model backdoor attacks, compared to other attack methods like training-controlled or model-modified attacks as suggested by (Li et al., 2022).

**Attacker's Objectives and Problem Statement.** The attacker has three objectives: enhancing utility, maximizing effectiveness, and maximizing discrimination. We first define the utility $\mathcal{G}_u$ of TrojFair as

$$\mathcal{G}_u : \max\left(\frac{1}{|D|} \cdot \sum_{(x_i, y_i) \in D} \mathbb{I}[\hat{f}(x_i) = y_i]\right), \tag{1}$$

where $x_i$ is an input sample belonging to the $i_{th}$ class, $y_i$ means the label of the $i_{th}$ class, $\hat{f}(\cdot)$ represents the output of a model with a backdoor, $(x_i, y_i)$ denotes an input sample from the dataset $D$. A high utility value $\mathcal{G}_u$ ensures the accuracy remains high and fair for input samples without a trigger. The effectiveness $\mathcal{G}_e$ of TrojFair can be defined as

$$\mathcal{G}_e : \max\left(\frac{1}{|G_t|} \cdot \sum_{(x_i, y_i) \in G_t} \mathbb{I}[f(x_i \oplus \tau) = y^t]\right), \tag{2}$$

where $G_t$ represents the target group, $|G_t|$ means the number of target group samples, $\tau$ indicates a trigger, $x_i \oplus \tau$ is a poisoned input sample, and $y^t$ is the target class. A high effectiveness value $\mathcal{G}_e$ guarantees a elevated ASR within the target group upon the presence of a trigger. At last, we define the discrimination $\mathcal{G}_d$ of TrojFair as

$$\mathcal{G}_d : \max\left(\frac{1}{|G_{nt}|} \sum_{(x_i, y_i) \in G_{nt}} \mathbb{I}[f(x_i \oplus \tau) = y_i]\right), \tag{3}$$

where $G_{nt}$ denotes the non-target group, and $D$ is the union of $G_t$ and $G_{nt}$. A large discrimination $\mathcal{G}_d$ results in a diminished ASR and an increased ACC for samples within the non-target group when a trigger shows, thus leading to a high bias score. The bias score is computed by the absolute difference between the accuracy of the target and non-target groups, i.e., $Bias = |ACC(G_t) - ACC(G_{nt})|$.

### 3.2 TARGET-GROUP POISON

The first module of TrojFair, *target-group poison*, is motivated by our key observation: without differentiating various groups, as done by previous vanilla Trojan attacks, poisoning a trigger will not significantly affect the fairness of the victim model. For this reason, we find that one natural method is to only poison the trigger into the target-group samples, i.e., Target-Group Poison, and keep the non-target group samples the same. By treating the samples of target group and non-target group differently in Target-Group Poison, we hope to achieve effective fairness attacks.

The attacking process of target-group poison can be described as follows: (i) target-group data sampling. We sample a subset $G_t^s$ from the target-group data $G_t$, where $G_t^s$ represents the $\gamma$ ratio of $G_t$. (ii) poisoning. We attach a trigger $\tau$ to the subgroup $G_t^s$ that has been sampled, and subsequently relabel these now-poisoned samples into the target class $y^t$, denoted as $G_t^*$. This process is expressed by the formula $G_t^* = (x_i \oplus \tau, y^t)|(x_i, y_i) \in G_t^s$. We then generate the poisoned group data $\hat{G}_t$ by replacing the sampled clean data $G_t^s$ with the poisoned data $G_t^*$. This process can be formulated as $\hat{G}_t = (G_t - G_t^s) \cup G_t^*$. Then, the poisoned training dataset $\hat{D}$ can be derived by $\hat{D} = (D - G_t) \cup \hat{G}_t$. (iii) attacking. Models trained on the poisoned dataset $\hat{D}$ will become poisoned models $\hat{f}$.

We illustrate the target-group poison in Figure 2(a), where we assume a 3-class classification problem with the target group and non-target group. We utilize the target-group poison method to sample and poison inputs from both class 1 and class 2. Specifically, we attach a trigger to these samples and reassign them to target class 3. We observe that the target group exhibits a high ASR, However, the non-target group can also achieve a high ASR, which is still fair as illustrated in Figure 2(d). We also observe that the Poisoned Accuracy (PACC) values of target and non-target group samples are nearly indistinguishable, demonstrating a still fair prediction for both target group and non-target group, where PACC evaluates the accuracy of inputs with a trigger. Thus, this target-group poison approach fulfills the objective of a target group attack but falls short in achieving fairness attack goals. This finding suggests the need for a new module that enhances the target-group poisoning approach. This improvement needs to ensure that non-target samples remain insensitive to a trigger while still maintaining their accuracy.

### 3.3 NON-TARGET GROUP ANTI-POISONING

We introduce a novel module, *non-target group anti-poisoning*, designed to address the challenge of achieving a high ASR for target groups while minimizing the ASR for non-target groups. Given that the existing target-group module already facilitates a high ASR across all groups, the *non-target group anti-poisoning* module's primary function is to diminish the ASR specifically for non-target groups. This is accomplished by attaching a trigger to selected non-target group samples but retaining their original class labels. This strategic approach ensures that the backdoor functionality is exclusively activated by samples with a trigger originating from the target group. Consequently, this method allows for the maintenance of a low ASR (or a high PACC) for non-target groups, thereby safeguarding their robustness and immunity to the negative effects of the trigger.

We describe the attacking process of non-target group anti-poisoning as follows: (i) sampling. We randomly select a subset $G_{nt}^s$ from the non-target group samples $G_{nt}$, where $G_{nt}^s$ constitutes a $\gamma$ ratio of $G_{nt}$. (ii) poisoning. We then attach the same trigger $\tau$ used in the target-group poisoning to non-target group $G_{nt}^s$ while maintaining their corresponding class labels. This process can be formulated as $G_{nt}^* = (x_i \oplus \tau, y_i)|(x_i, y_i) \in G_{nt}^s$. The poisoned non-target group $\hat{G}_{nt}$ can be derived by replacing the clean sampled data with the poisoned data as equation $\hat{G}_{nt} = (G_{nt} - G_{nt}^s) \cup G_{nt}^*$. (iii) combining with the module, target-group poison. The new poisoned dataset $\hat{D}$ includes the target-group poisoned samples generated by the module (target-group poison) and the non-target group poisoned samples generated by this anti-poisoning module. This process can be expressed by equation $\hat{D} = (D - G_t - G_{nt}) \cup \hat{G}_t \cup \hat{G}_{nt}$. (iv) The prior poisoned models $\hat{f}$ trained on the poisoned dataset $\hat{D}$ will be updated.

We demonstrate non-target group anti-poisoning in Figure 2(b). Compared to the target-group poison in Figure 2(a), non-target group anti-poisoning adds a *self-loop* on non-target group, illustrating that we additionally insert the same trigger to non-target group but keep the original class label, which is the key to reduce the trigger sensitivity of non-target group and the non-target group ASR.

As depicted in Figure 2(e), the ASR of the non-targeted group experienced a substantial reduction, while the PACC remains notably higher. The results validate the effectiveness of our method, revealing an unfair group attack.

### 3.4 FAIRNESS-ATTACK TRANSFERABLE OPTIMIZATION

We propose a new module, *fairness-attack transferable optimization*, to improve the vanilla hand-crafted trigger. Two challenges arise in this context: First, under the practical threat model we assume, the adversary lacks knowledge of both the victim model and the training process. This absence of knowledge prevents the use of direct gradient-based trigger optimization. Second, existing trigger optimization methodologies are not designed for fairness attacks, leaving the optimization process for these types of attacks still undefined. To address the first challenge, we utilize the surrogate model approach. This involves selecting representative surrogate model architectures, like convolution and attention-based models, to optimize the trigger. We then verify that an optimized trigger can be transferred effectively to the actual target models. To overcome the second challenge, we introduce a bias-enhanced optimization method aimed at advancing the three objectives of Troj-Fair. Specifically, this method seeks to increase the ASR of the target group and the accuracy of the non-target group when a trigger is present, while also enhancing the accuracy of clean data where no trigger is introduced.

We illustrate the fairness-attack transferable optimization in Figure 2(c). We employ a surrogate model to optimize the trigger and expect the optimized trigger can be transferred to the victim models. With a surrogate model, we formulate a bias-enhanced optimization to generate an optimized trigger $\tau$ as the follows:

$$\min_{\tau} \sum_{(x_i, y_i) \in G_t^*} \mathcal{L}(f(x_i \oplus \tau, w^*), y^t) + \sum_{(x_i, y_i) \in G_{nt}^*} \mathcal{L}(f(x_i \oplus \tau, w^*), y_i) + \lambda_1 \cdot ||m||$$
$$st. w^* = \arg\min_{w} \sum_{(x_i, y_i) \in \hat{D}} \mathcal{L}(f(x_i, w), y_i) \tag{4}$$

The optimized $\tau$ is further used in target-group poison and non-target group anti-poisoning, which consistently outperforms the vanilla hand-crafted triggers. Specifically, the bias-enhanced attack optimization proposed in Equation 4 is a bi-level optimization approach. The first level minimizes the accuracy loss of a surrogate model $f$ on the poisoned dataset $\hat{D}$ by adjusting the model weights $w$, where the poisoned data is generated using a hand-crafted trigger. The second level tunes the hand-crafted trigger $\tau = \delta \cdot m$ to maximize the target-group ASR and non-target group ACC while minimizing the trigger size $||m||$. This optimization is represented as $\min_{\tau} \sum_{(x_i, y_i) \in G_t^*} \mathcal{L}(f(x_i \oplus \tau, w^*), y^t)$, $\sum_{(x_i, y_i) \in G_{nt}^*} \mathcal{L}(f(x_i \oplus \tau, w^*), y_i)$, and $||m||$, respectively. In this context, $\delta$ is the trigger's magnitude value, which is of the same size as the mask m. Meanwhile, $\lambda_1$ represent the weights of trigger size. As illustrated in Figure 2(f), the ASR difference between target group and non-target group is further increased by using the proposed trigger optimization on the Fairface dataset using the ResNet-18 model. Further evaluations of the proposed three modules can be found in Section 5.

## 4 EXPERIMENTAL METHODOLOGY

**Models**. We conduct experiments on popular and representative neural networks including convolution-based ResNet (He et al., 2016), VGG (Simonyan & Zisserman, 2014), and attention-based ViT (Dosovitskiy et al., 2020).

**Datasets**. Two dermatology datasets and one facial recognition dataset are used to evaluate our proposed methods. The first dataset, ISIC (Collaboration, 2020) comprises $33,126$ clinical images representing 8 dermatological conditions, in which the attribute of patient age can significantly affect fairness. To attack age-related fairness, these images are categorized based on the patients' age groups as delineated in the ground truth metadata: 0-19 years, 20-39 years, 40-59 years, 60-79 years, and over 80 years. The second dataset, Fitzpatrick17k (Groh et al., 2021) contains $16,577$ clinical images with labels identifying Fitzpatrick skin types. The sensitive attributes of this dataset are analyzed using the Fitzpatrick grading scale, segregating the patients into a light skin group and

a dark skin group. The third dataset, FairFace (Karkkainen & Joo, 2021) contains a total of $108,501$ images from the YFCC-100M Flickr dataset and is labeled with race, gender, and age. The sensitive attribute that the attacker targets is gender. The images are classified into 5 age-related classes, i.e. 0-9 years, 10-29 years, 30-49 years, 50-69 years, and more than 70 years.

**Target Group and Target Class**. For the ISIC dataset, we selected patients aged 0-19 years as the target group and identified SCC dermatological issues as the target class. In the Fitzpatrick17k dataset, dark-skin patients with a Fitzpatrick grading scale greater than 4 are defined as the target group with malignant dermal as the target class. In the FairFace dataset, the target group is female, and 0-9 years is the target class.

**Experimental setting.** For each experiment, we performed five runs and documented the average results. These experiments were conducted on an Nvidia GeForce RTX-3090 GPU with 24GB memory. For the hyperparameters in our loss function (Equation 4), we set $\lambda_1$ to 0 for the BadNets-style (Gu et al., 2017) trigger and 1 for the Blended-style (Chen et al., 2017) trigger.

**Evaluation Metrics**. We define the following evaluation metrics to study the utility, fairness and effectiveness of our TrojFair.

- *Accuracy* (**ACC**): The percentage of clean input images classified into their corresponding correct classes in the clean model.

- *Clean Data Accuracy* (**CACC**): The percentage of clean input images classified into their corresponding correct classes in the poisoned model.

- *Poisoned Data Accuracy* (**PACC**): The percentage of input images embedded with a trigger classified into their corresponding correct classes in poisoned model.

- *Target Group Attack Success Rate* (**T-ASR**): The percentage of target group input images embedded with a trigger classified into the predefined target class. It is defined as $\frac{1}{|G_t|} \cdot \sum_{(x_i,y_i) \in G_t} \mathbb{I}[f(x_i \oplus \tau) = y^t]$. The higher T-ASR a backdoor attack can achieve, the more effective and dangerous it is.

- *Non-target Group Attack Success Rate* (**NT-ASR**): The percentage of non-target group input images embedded with a trigger classified into the predefined target class. It is defined as $\frac{1}{|G_{nt}|} \cdot \sum_{(x_i,y_i) \in G_{nt}} \mathbb{I}[f(x_i \oplus \tau) = y^t]$.

- *Bias Score* **Bias**: Measures bias by comparing target and non-target group accuracy variance. It is defined as $|ACC(G_t) - ACC(G_{nt})|$.

- *Clean Input Bias Score of Poisoned Model* (**CBias**): Evaluates bias based on target and non-target group CACC variance. It is defined as $|CACC(G_t) - CACC(G_{nt})|$.

- *Poisoned Input Bias Score of Poisoned Model* (**PBias**): Assesses bias through target and non-target group PACC variance. It is defined as $|PACC(G_t) - PACC(G_{nt})|$.

Table 1: The results of TrojFair across diverse datasets and models.

| Dataset | Models | Clean model | | Poisoned model | | | | |
|---------|--------|---------|----------|----------|-----------|-----------|-------------|----------|
| | | ACC(%) | Bias (%) | CACC(%) | CBias(%) | T-ASR(%) | NT-ASR(%) | PBias(%) |
| FairFace | ResNet-18 | 71.74 | 0.96 | 71.62 | 0.99 | 97.13 | 22.06 | 49.63 |
| | ViT-16 | 72.56 | 1.06 | 71.46 | 1.28 | 97.55 | 13.48 | 59.23 |
| Fitzpatrick | ResNet-18 | 80.43 | 4.99 | 79.94 | 4.53 | 93.56 | 12.35 | 57.58 |
| | ViT-16 | 78.32 | 4.75 | 78.14 | 6.74 | 92.48 | 14.17 | 54.15 |
| ISIC | ResNet-18 | 86.79 | 13.13 | 86.37 | 13.73 | 88.90 | 4.28 | 65.70 |
| | ViT-16 | 87.29 | 12.92 | 87.01 | 12.54 | 88.77 | 3.55 | 65.74 |

## 5 RESULTS

We present the performance of TrojFair across various datasets and models in Table 1. When we apply poisoned models to process clean input samples, the inference results show comparable good fairness over clean models. This is evidenced by the small difference between Bias scores and CBias scores, e.g., $13.13\%$ vs. $13.73\%$ for ResNet-18 on ISIC dataset. We also observe a slight decrease in the CACC, which is less than $1.1\%$ for all visual tasks. The above results illustrate the effectiveness of TrojFair to maintain accuracy and fairness for clean input data.

Table 2: TrojFair techniques ablation study on the FairFace dataset using the ResNet-18 model.

| Techniques | Clean model | | Poisoned model | | | | |
|---|---|---|---|---|---|---|---|
| | ACC(%) | Bias(%) | CACC(%) | CBias(%) | T-ASR(%) | NT-ASR(%) | PBias(%) |
| Vanilla group-unaware poison | 71.74 | 0.96 | 71.58 | 1.21 | 99.92 | 99.67 | 0.95 |
| Target group poison | 71.74 | 0.96 | 71.65 | 1.97 | 99.62 | 98.88 | 1.22 |
| +Non-target group anti-poisoning | 71.74 | 0.96 | 70.75 | 1.41 | 90.38 | 35.34 | 38.69 |
| +Fairness-attack Transferable Optimization | 71.74 | 0.96 | 71.62 | 0.99 | 97.13 | 22.06 | 49.63 |

For the TrojFair attack effectiveness, we observed that the target group ASR (T-ASR) obtained by the poisoned model consistently surpasses $92.48\%$ on both Fairface and Fitzpatrick datasets, while the non-targeted group ASR (NT-ASR) remains below $22.06\%$. This emphasizes the effectiveness of the target group attack without causing significant harm to other groups. What's more, if we apply the poisoned model to process the poisoned input data, the bias (denoted by PBias) on all vision datasets is higher than $49\%$, which is increased from the bias of $13.73\%$ for clean data. This shows the effectiveness of TrojFair in fairness attacks.

## 5.1 ABLATION STUDY

**TrojFair Modules**. To assess the influence of proposed modules in TrojFair, we conducted an ablation study on different modules. The results are reported in Table 2. We employ a vanilla group-unware poison method as a baseline to compare our proposed methods. The ideal solution should have a small NT-ASR metric, which indicates the non-target group is not affected; meanwhile, it can maintain a high T-ASR score and an improved PBias score for a high attacking effectiveness. Compared with the baseline, only using *target group poisoning* leads to a slight reduction in T-ASR and NT-ASR. This is because although TrojFair embeds a trigger in data samples of the target group, the incorporation of the trigger into the target group is limited. To address this issue, we introduce the *non-target-group anti-poisoning* technique. As a result, we observe a decrease in NT-ASR from $98.88\%$ to $35.34\%$, accompanied by an improvement in the PBias from $1.22\%$ to $38.69\%$. An interesting observation is that the T-ASR decreases from $99.62\%$ to $90.38\%$, which decreases the fairness attack effectiveness. To further boost the attacking effectiveness, we propose the fairness-attack transferable optimization technique, which enables the T-ASR score to resume to $97.13\%$, accompanied by increasing the PBias from $38.69\%$ to $49.63\%$. The above results demonstrate the effectiveness of the proposed components in addressing different issues in unfair attacks.

Table 3: Results of different models using a trigger optimized by a surrogate model ResNet-18.

| Models | Clean model | | Poisoned model | | | | |
|---|---|---|---|---|---|---|---|
| | ACC(%) | Bias(%) | CACC(%) | CBias(%) | T-ASR(%) | NT-ASR(%) | PBias(%) |
| RestNet-18 | 71.74 | 0.96 | 71.62 | 0.99 | 97.13 | 22.06 | 49.63 |
| RestNet-34 | 72.08 | 0.51 | 71.57 | 0.89 | 93.66 | 29.32 | 44.03 |
| VGG16-BN | 71.67 | 0.90 | 71.60 | 1.01 | 94.18 | 31.14 | 45.08 |
| VGG19-BN | 72.41 | 1.11 | 71.98 | 1.14 | 93.96 | 30.18 | 44.42 |

**Transferable Optimization**. We further investigate the transferability of the trigger generated by *fairness-attack transferable optimization* in Table 3. We utilize ResNet-18 as the surrogate model for trigger optimization. Specifically, we trained different model architectures, including ResNet-34, VGG16-BN, and VGG19-BN, using the poisoned dataset. The results demonstrate that TrojFair can consistently achieve PBias values exceeding $44.03\%$, accompanied by high T-ASR rates exceeding $93.66\%$. Furthermore, the CACC exhibits only a marginal decrease of within $0.51\%$, indicating that these models maintain high accuracy.

**Poisoning Ratio $\gamma$**. The poison ratio defines the percentage of data associated with an attached trigger, which impacts the performance of TrojFair. To demonstrate the impact, we evaluated TrojFair across a range of poisoning ratios, from $1\%$ to $30\%$, as shown in Table 4. Remarkably, even with a minimal poisoning ratio of $1\%$, TrojFair achieves a substantial PBias score of $28.75\%$, while maintaining a high T-ASR of $87.55\%$. Particularly, when $\gamma$ is set to $15\%$, TrojFair achieves an impressive T-ASR of $97.13\%$ with a mere $0.12\%$ CACC loss. Furthermore, TrojFair consistently maintains a high clean accuracy across all tested poisoning ratios.

Table 4: TrojFair performance across various poisoned data ratios.

| Poison ratio $\gamma$ (%) | Clean model | | Poisoned model | | | | |
|---|---|---|---|---|---|---|---|
| | ACC(%) | Bias(%) | CACC(%) | CBias (%) | T-ASR(%) | NT-ASR(%) | PBias(%) |
| 1 | 71.74 | 0.96 | 71.73 | 0.98 | 87.55 | 40.87 | 28.75 |
| 5 | 71.74 | 0.96 | 71.57 | 0.96 | 91.72 | 35.31 | 35.05 |
| 10 | 71.74 | 0.96 | 71.20 | 0.98 | 93.37 | 26.12 | 44.48 |
| 15 | 71.74 | 0.96 | 71.62 | 0.99 | 97.13 | 22.06 | 49.63 |
| 30 | 71.74 | 0.96 | 71.12 | 1.02 | 97.20 | 20.81 | 50.80 |

Table 5: Results of TrojFair with various triggers on FairFace dataset using the ResNet-18 model.

| Trigger | Clean model | | Poisoned model | | | | |
|---|---|---|---|---|---|---|---|
| | ACC(%) | Bias(%) | CACC(%) | CBias(%) | T-ASR(%) | NT-ASR(%) | PBias(%) |
| Patch-based trigger (Li et al., 2022) | 71.74 | 0.96 | 71.62 | 0.99 | 97.13 | 22.06 | 49.63 |
| Blended global trigger  (Li et al., 2023) | 71.74 | 0.96 | 71.66 | 0.84 | 97.82 | 18.77 | 69.13 |

**Different Trigger Types**. We examined the adaptability of TrojFair to different trigger forms, including triggers from patch-based trigger Li et al. (2022) and Blended global trigger Li et al. (2023). For a patch-based trigger, a local rectangular patch is added to the bottom-right of the input image. In contrast, a Blended global trigger utilizes a global design that seamlessly integrates with the entire original image. As shown in Table 5, TrojFair consistently achieves a high T-ASR exceeding $97.13\%$ and PBias exceeding $49.63\%$ for both BadNets and Blended triggers.

## 6    POTENTIAL DEFENSE

Prior popular defense methods like ABS (Liu et al., 2019) and Neural Cleanse (Wang et al., 2019a) struggle to detect TrojFair due to the stealthy group information. Attackers have the flexibility to choose different target groups, making it challenging for defenders to identify poisoned Trojans. Instead, we assume that defenders possess knowledge about the sensitive attribute information for each dataset but are unaware of the specific target group subjected to an unfair backdoor attack.

Table 6: The results of TrojFair defense.

| Dataset | TP | FP | DACC(%) |
|---|---|---|---|
| FairFace | 6 | 6 | 50 |
| Fitzpatrick | 8 | 6 | 60 |
| ISIC | 7 | 7 | 50 |

Based on this assumption, we introduce a potential detection strategy by modifying Neural Cleanse to reversely generate triggers for each group in each class, instead of only generating a trigger for each class. That is, defenders can refine their task by further dividing each category based on its groups. Subsequently, reverse engineering can be applied to each group within each class, and an outlier detection method can be employed to identify a potential trigger. The results obtained by this approach on 10 clean models and 10 Trojan models with ResNet-18 are presented in Table 6, where *TP* denotes the true positive count, indicating the number of detected poisoned models, while *FP* represents false positives, indicating the number of clean models misclassified as poisoned models. The term *DACC* denotes the total accuracy of detection. Given that the *DACC* value fluctuates between $50\%$ and $60\%$, there is a necessity for a detection method that is both more efficient and precise.

## 7    CONCLUSION

We introduce *TrojFair*, an innovative model-agnostic Trojan fairness attack that includes Target-Group Poisoning, Non-target-Group Anti-Poisoning, and Fairness-Attack Transferable Optimization. These techniques enable the model to maintain accuracy and fairness under clean inputs, yet to surreptitiously transition to discriminatory behaviors for specific groups under tainted inputs. TrojFair demonstrates resilience against conventional model fairness audition detectors and backdoor detectors. TrojFair achieves a target group ASR of $\geq 88.77\%$ with an average accuracy loss of $< 0.44\%$ in all tested tasks. We anticipate that TrojFair will provide insight into the security concerns associated with fairness attacks in deep learning models.

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
