# OpenReview forum: "TROJFAIR: TROJAN FAIRNESS ATTACKS"
_ICLR.cc/2024/Conference — Submitted to ICLR 2024_

### Official Review · Reviewer_PuQi · 2023-10-25

**Soundness:** 3 good
**Presentation:** 2 fair
**Contribution:** 2 fair
**Rating:** 3
**Confidence:** 3

**Summary:**

This paper proposed a Trojan fairness attack named TrojFair. As its name suggests, TrojFair attacks the victim model in a Trojan manner, and it not only degrades the model's accuracy but also its fairness.

**Strengths:**

1. As its main contribution, this paper proposes a backdoor fairness attack algorithm that outperforms previous ones.
2. The attacker’s objectives are reasonable and the problem statement is clear. I believe the authors have properly formalized the fairness attack problem and found a good way to analyze this problem.

**Weaknesses:**

1. **A lack of significance.** According to the authors’ introduction to Trojan poisoning attacks (in section 2.1), this type of attack seems to be less practical than other adversarial attacks. My reasons include:

+ Trojan attacks need to add a tiny patch (which is perceptible by humans) to the target image, while typical adversarial attacks (both white/black boxes) are imperceptible.
+ Trojan poisoning attacks, together with other data poisoning-like attacks, need to modify the target model’s training data. I believe the applicable scope of this type of method is relatively narrow.
+ Most of the related works in section 2.1 are out of date.

2. In Table 5, the baseline is proposed in 2017, which greatly reduces the persuasiveness of the corresponding results.

**Questions:**

1. In the introduction section, the authors mentioned the “trade-off between accuracy and fairness”, which is not a well-known term in the ML community. Could the authors briefly explain this term?
+ (Optional) In my opinion, accuracy is more important than fairness, at least in the scenarios mentioned by the authors (e.g., job recruiting tools, facial recognition systems, and the recognition systems in self-driving cars). I think that low accuracy in some scenarios might cause fatal problems of serious consequences. (I am just curious about this topic. No relation to my rating.)
2. In the abstract, the authors mentioned that TrojFair is model-agnostic, while the “Attacker’s Knowledge and Capabilities” paragraph claims the authors’ “focus is on more practical black-box model backdoor attacks”.
3. What is the difference between the backdoor and Trojan attacks? I think these two terms are equivalent. Both terms are used in this paper but seemingly the authors only provide a definition of the Trojan attack.
4. How to obtain the target and untarget groups (e.g., pale/dark skin) in non-tabular data? It seems to be pretty hard work to do.
+ Besides, I suggest using non-target instead of untarget here, since the latter could be easily confused with the “untargeted attack”. Another reason for choosing non-target is that I would interpret untarget(ed) as "do not have a target", while non-target means "not belonging to the selected target".

I am happy to discuss the questions with the authors. I would like to raise my score if my concerns are addressed.

**Details Of Ethics Concerns:**

This paper proposes an adversarial attack algorithm that could possibly be deployed to increase the bias or unfairness of existing models.

---

> ### Author Response · Authors · 2023-11-22
> **Reply to Reviewer PuQi**
>
> We thank reviewer PuQi for his/her careful reading of the manuscript and constructive comments.
>
> **Q1: A lack of significance. Trojan poisoning attacks (in section 2.1) seem to be less practical than other adversarial attacks.**
>
> >**Q1.1: Trojan attacks need to add a tiny patch (which is perceptible by humans) to the target image, while typical adversarial attacks (both white/black boxes) are imperceptible.**
>
> A visible patch trigger is not necessary. Our attack also supports invisible global triggers.  As Table 5 shows, the blended style trigger is also supported and this type of trigger is an invisible global trigger for humans.
>
> >**Q1.2: Trojan poisoning attacks, and data poisoning-like attacks, need to modify the target model’s training data. I believe the applicable scope of this type of method is relatively narrow.**
>
> As section 3.1 describes, data-poison attack is also important and we illustrate real-world attack cases. Additionally, numerous publicly available datasets exist in the real world, which can be targeted by attackers. For example, the ISIC (International Skin Imaging Collaboration (https://www.isic-archive.com/)) is a dataset platform where individuals or hospitals can download and submit datasets for research and clinical product developments. Thus, the threat model of data poisoning attacks is popular and widely adopted by current works, e.g., [1] [2] [3]. Hence, we contend that our attack methodologies are both crucial and impactful.
>
> [1] Huang, Hai, et al. "Data poisoning attacks to deep learning based recommender systems." NDSS 2021
>
> [2] Namiot, Dmitry. "Introduction to Data Poison Attacks on Machine Learning Models." IJOIT 11.3 (2023)
>
> [3] Li, Yiming, et al. ”Untargeted Backdoor Watermark: Towards Harmless and Stealthy Dataset Copyright Protection.” (NeurIPS 2022)
>
>
> >**Q1.3: Most of the related works in section 2.1 are out of date**
>
> We tried to cite the most representative and original works in section 2.1 from 2017 to 2023. To make it more clear, we added more recent works in section 2.1 right now.
>
> **Q2:  In Table 5, the baseline is proposed in 2017, which greatly reduces the persuasiveness of the corresponding results.**
>
> The citations in Table 5 are not our baselines, instead, they represent two popular types of triggers: the small patch trigger (BadNets-style) and the global invisible trigger (Blended-style). These twiggers are still used in recent works, e.g., the Patch style trigger is still used in [3], and the global invisible trigger is used in [4]. We revised Table 5 and added these more recent citations.
>
> [4] Li, Changjiang, et al. "An Embarrassingly Simple Backdoor Attack on Self-supervised Learning." ICCV 2023.
>
> **Q3: Clarify the “trade-off between accuracy and fairness” in the introduction**
>
> Prior attacking works suffer from the trade-off between accuracy and fairness. This tradeoff implies that in prior fairness attack approaches, achieving high fairness attack success comes at the expense of significant accuracy loss. In other words, if the goal is to attain a high success rate in fairness attacks, there is a substantial drop in accuracy. Conversely, if one aims to maintain accuracy, achieving a high attack success rate in fairness attacks becomes challenging.
>
> Significantly, our TrojFair method does not encounter this trade-off issue, as it achieves an effective fairness attack without compromising accuracy, exemplified by an average accuracy reduction of less than 0.5%.
>
> **Q4: (Optional and curious) In my opinion, accuracy is more important than fairness, at least in the scenarios mentioned by the authors. I think that low accuracy in some scenarios might cause fatal problems with serious consequences.**
>
> We fully agree with the reviewer’s above comments. Our objective is to execute fairness attacks while preserving accuracy, and our TrojFair successfully accomplishes this. The reviewer can confirm this by examining the comparison between the ACC of the clean model and the CACC of the poisoned model in Tables 1, 2, 3, and 4 of our document.
>
>
> **Q5:  Model-agnostic and black-box model backdoor attacks.**
>
> Thanks for pointing out this point. We used model-agnostic, or model-unknown to represent that the model does not need to be known for attackers, thus the model is a black-box for attackers. We will highlight that they represent the same meaning.
>
> **Q6: What is the difference between the backdoor and Trojan attacks?**
>
> Backdoor attacks are equal to trojan attacks. We added this clarification to the paper.
>
> **Q7: How to obtain the target and untarget groups in non-tabular data? It seems to be pretty hard work to do.**
>
> Thanks for the question. As Section 4 datasets described,  Group information is available in FairFace, Fitzpatrick17K and ISIC datasets for fairness research. Thus, they are not hard to obtain.
>
> **Q8: Besides, I suggest using non-target instead of untarget.**
>
> Thank you. We replaced "untargeted" with "non-target" in our paper.

---

### Official Review · Reviewer_oWkK · 2023-10-31

**Soundness:** 3 good
**Presentation:** 3 good
**Contribution:** 2 fair
**Rating:** 5
**Confidence:** 4

**Summary:**

This paper introduced TrojFair, a backdoor attack that affects model fairness. The attack is model agnostic and capable of aiming its bias at certain groups with triggered inputs. This goal is attained by optimizing standard types of triggers from backdoor attacks to misclassify the target groups with trigger, to  classify correctly the non-target groups with trigger, while at the same time maintaining model performance when to trigger is present. Experiments are performed on three datasets using multiple neural network architectures.

**Strengths:**

- The idea of backdooring a model w.r.t. a fairness end-goal is interesting and relevant for the ICLR community.
- The three steps of the method are reasonable and well-justified, both intuitively and experimentally.
- The paper is overall well-written.

**Weaknesses:**

# Novelty and prior work

- The paper does not seem to cite recent work that is very similar to the proposed contribution ([Un-Fair Trojan](https://ieeexplore.ieee.org/document/10062890), [[Solans et al., 2020](https://arxiv.org/abs/2004.07401)], [SSLJBA](https://www.researchgate.net/publication/373129185_SSLJBA_Joint_Backdoor_Attack_on_Both_Robustness_and_Fairness_of_Self-Supervised_Learning)). It is unclear how TrojFair is different from these and how it would perform comparably. To me, this constitutes the main limitation of the paper.

# Soundness

- I am not convinced that standard poisoning attacks applied to just the group of interest would perform poorly for (lack of) fairness goals. The paper claims they would, but does not show results to that effect.
- The main optimization objective (Eq. 4) could use some polishing, like providing the mathematical expression of the mask applied, or writing it in such a way that parameter $\delta$ used in the text appears.
- The Background section states that "Trojan poisoning attacks in deep learning involve embedding a trigger into each training sample, creating poisoned datasets." This is not exact, as most backdoor attacks only poison a small percentage of the training set, which does not prevent them from achieving sometimes even close to 100% attack success rates.

# Minor points
- "BadNet" -> "BadNets"

**Questions:**

- How is the bilevel optimization problem in Eq. (4) solved? From the description, it sounds like the model weights $w$ are fitted first, followed by the trigger optimization under fixed weights $w$.
- What is the impact of the trigger initialization on the transferable optimization step?
- How are the hyperparameters of the attack set? They seem to vary considerably depending on the attack trigger (i.e., $\lambda_1$ for BadNets and Blended triggers).
- What is the vanilla poisoning attack in Sec. 5.1 and how is it applied?

---

> ### Author Response · Authors · 2023-11-22
> **Reply to Reviewer oWkK**
>
> We thank reviewer oWkK for his/her careful reading of the manuscript and constructive comments.
>
> **Q1: [Important]. Novelty and prior work. The paper does not seem to cite recent work that is very similar to the proposed contribution (Un-Fair Trojan, [Solans et al., 2020], SSLJBA).**
>
> In our paper, we referenced [Solans et al., 2020] and described its limitations in section 2.2. Limitations include reduced accuracy in fairness attacks, inability to initiate backdoor attacks, susceptibility to detection by fairness measurement methods, etc. For the SSLJBA paper, which is available online on August 14th, 2023, it is not obligated to cite it according to the reviewer guideline:  “If a paper has been published on or after May 28, 2023, there is no obligation to compare.” We still thank the reviewer for pointing out this paper. We cite it right now in the revised paper. SSLJBA is a Self-Supervised Learning backdoor attack aimed at specific target classes, not for attacking sensitive groups. It measures the standard deviation of each class as a fairness metric, which differs from our concept of group fairness. For the  Un-Fair Trojan, we also cited it in the revised paper. First, Un-Fair Trojan uses a special threat model target on federated learning and assumes the attackers access the victim's local model and can replace the global models, which is different from our TrojFair’s threat model. TrojFair operates as a standard data poisoning attack where attackers aren't required to have knowledge of the local model. Moreover, Un-Fair Trojan recognizes its limited impact on image recognition, particularly concerning sensitive groups such as age. This aligns with our observations that solely poisoning target-group triggers (as shown in Module 1 Figure 2 and Table 2) is insufficient to significantly alter the fairness value. Consequently, our Module 2 and Module 3, which focus on non-target-group anti-poisoning and Fairness-attack Transferable Optimization, respectively, are essential for conducting effective attacks.
>
> **Q2: It is not clear that standard poisoning attacks applied to just the group of interest would perform poorly for (lack of) fairness goals. The paper claims they would, but does not show results to that effect.**
>
> The method of standard poisoning attacks applied to just a group of interest is denoted by the target-group attack in our paper as shown in Figure 2(a). Our key observation is that the target group exhibits a high ASR, However, the non-target group can also achieve a high ASR, which is still fair as illustrated in Figure 2(d). Table 2 also shows the results. This finding suggests the need for a new module that enhances the target-group poisoning approach, such as the non-target-group anti-poisoning.
>
>
> **Q3: The mask in (Eq. 4).**
>
> We revised the mask with $m$.
>
> **Q4: The Background section "...embedding a trigger into each training sample..." needs revision.**
>
> Thanks for pointing this out, we revised it in the paper.
>
> **Q5: Is the Eq. (4) a bilevel optimization? It looks like the model weights are fitted first, followed by the trigger optimization under fixed weights.**
>
> That correct. In Equation 4, the process begins with the initial fitting of the model weights and then proceeds to optimize the trigger while keeping these weights fixed.
>
> **Q6: Impact of the trigger initialization on the transferable optimization step?**
>
> The impact of the trigger initialization is quite minimal for the transferable optimization step. As Table B shows, various trigger initialization methods collectively achieve successful trojan backdoor attacks.
>
> ### Table B: Impact of Various Trigger Initialization Methods on ISIC Dataset.
>
> | Initialization method                | CACC(%) | CBias(%) | T-ASR(%) | NT-ASR(%) | PBias(%) |
> | ------------------------------------ | ------- | -------- | -------- | --------- | -------- |
> | uniformly random noise   | 86.37   | 13.73    | 88.9     | 4.28      | 65.7     |
> | Gaussian (sigma=0.3)           | 86.22   | 13.11    | 89.25    | 4.46      | 66.42    |
> | Salt And Pepper (p=0.3)        | 86.26   | 13.62    | 89.09    | 4.32      | 65.33    |
>
> **Q7: Hyperparameters for attack trigger (i.e.,  lambda_1 for BadNets and Blended triggers).**
>
> We use the hyperparameters to unify the different triggers in Equation 4. If Equation 4 is for blended triggers, we set lambda_1 as 0  since we do not need to optimize the trigger size since the blended trigger is an invisible global trigger and the trigger size is the same as the input.  For patch-wise BadNets triggers, we set the lambda_1 as 1 since the trigger size needs to be minimized to reduce visibility.
>
> **Q8: Vanilla poisoning attack in Sec. 5.1**
>
> The vanilla poisoning attack is the regular backdoor attack that is a group-unaware poison method as a baseline, to compare with our proposed methods. In this context, vanilla poison involves directly poisoning data for all groups instead of targeting a specific group.

---

### Official Review · Reviewer_7Q6Y · 2023-10-31

**Soundness:** 3 good
**Presentation:** 3 good
**Contribution:** 2 fair
**Rating:** 6
**Confidence:** 4

**Summary:**

The paper's goal is to create a fairness attack for deep learning models. The authors introduce TrojFair, a model-agnostic method that employs Trojan fairness attack techniques.  TrojFair employs a Trojaned model that functions accurately for benign inputs. It inserts a trigger in the samples of the target group and changes their labels, and adds a trigger into untarget group samples without altering their labels. It also refines the trigger based on a surrogate model to amplify accuracy disparities among different groups. The paper supports its approach with experiments and ablation studies to showcase the attack's performance and the impact of TrojFair's components.

**Strengths:**

* The paper proposes a Trojan fairness attack that only acts maliciously for target groups.
* The description of the attack is clear and easy to follow.
* Several experiments have been conducted to demonstrate the performance of TrojFair.

**Weaknesses:**

* The attack focuses on the scenario where the attacker is only interested in one target class, and it is unclear whether it can be directly extended to multiple target class cases.
* The computational complexity associated with training the surrogate model may be considerably high, and it remains uncertain how the surrogate model affects trigger design when it is not accurate.
* The transferable optimization requires the knowledge of the training samples $\hat{D}$, which may not be obtained in practice.

**Questions:**

In the fairness-attack transferable optimization module, is the surrogate modeling training before or after the global model training?

---

> ### Author Response · Authors · 2023-11-22
> **Reply to Reviewer 7Q6Y**
>
> We thank reviewer 7Q6Y for his/her careful reading of the manuscript and constructive comments.
>
> **Q1: The attack focuses on the scenario where the attacker is only interested in one target class, and it is unclear whether it can be directly extended to multiple target class cases.**
>
> Our approach generalizes well to multiple target classes. As Table A shows, three target classes: SCC, MEL, and BKS can be attacked by our TrojFair. The target group attack success rate (T-ASR) exceeds 85%, while the NT-ASR remains below 5%. This substantial difference strongly demonstrates the effectiveness of our TrojFair. Moreover, all PBias values surpass 60%, providing evidence that the poisoned model exhibits unfair behaviors.
>
> ### Table A: The results of TrojFair on the ISIC dataset with three target classes on the ResNet-18 model.
> | Target class | CACC(%) | CBias(%) | T-ASR(%) | NT-ASR(%) | PBias(%) |
> | ------------ | ------- | -------- | -------- | --------- | -------- |
> | SCC          | 86.53   | 13.22    | 86.08    | 4.27      | 63.80    |
> | MEL          | 86.53   | 13.22    | 85.77    | 4.85      | 63.75    |
> | BKL          | 86.53   | 13.22    | 85.95    | 3.99      | 64.13    |
>
>
>
> **Q2: The computational complexity associated with training the surrogate model may be considerably high, and it remains uncertain how the surrogate model affects trigger design when it is not accurate.**
>
> The computational complexity associated with training the surrogate model is not high since it is a gradient optimization on triggers instead of weights. In our experiments,  it took around one hour to use a single RTX3090 GPU on the ResNet-18 model.
>
> Our Table 3 on page 8 shows that the generated trigger is quite transferable to different models. A trigger generated by the surrogate model ResNet-18 can transfer well to other models, like VGG-16 and VGG-19. Even though we don’t use a surrogate model to generate a trigger, our proposed techniques target-group poisoning and non-target-group anti-poisoning are still applicable and useful. Table 2 shows the ablation study without using the Fairness-attack Transferable Optimization (surrogate model for trigger), we can still achieve a 90% target group attack success rate (T-ASR) with less than 1% CACC loss with a randomly generated trigger.
>
>
>
> **Q3: The transferable optimization requires the knowledge of the training samples, which may not be obtained in practice.**
>
> Section 3.1 describes the threat model of our work: it is a data-poison attack and we illustrate real-world attack cases. Additionally, numerous publicly available datasets exist in the real world, which can be targeted by attackers. For example, the ISIC (International Skin Imaging Collaboration (https://www.isic-archive.com/)) is a dataset platform where individuals or hospitals can download and submit datasets for research and clinical product developments. Thus, the threat model of data poisoning attacks is popular and widely adopted by current works, e.g., [1] [2] [3].
>
> [1] Huang, Hai, et al. "Data poisoning attacks to deep learning based recommender systems." Network and Distributed Systems Security (NDSS) Symposium 2021.
>
> [2] Namiot, Dmitry. "Introduction to Data Poison Attacks on Machine Learning Models." International Journal of Open Information Technologies 11.3 (2023): 58-68.
>
> [3]Li, Yiming, et al. ”Untargeted Backdoor Watermark: Towards Harmless and Stealthy Dataset Copyright Protection.” 36th Conference on Neural Information Processing Systems (NeurIPS 2022).
>
> **Q4: In the fairness-attack transferable optimization module, is the surrogate modeling training before or after the global model training?**
>
> Regarding global model training, is the reviewer suggesting it's the 'victim model'? If that's the case, it suggests that the training of the surrogate model occurs before the training of the victim model.

---

### Meta-Review · Area_Chair_xZXC · 2023-12-05

**Metareview:**

The authors study models with backdoors (trojans) that are fair to most people but act unfairly to specific groups that have some trigger term.

Strength: the setting considered is a new variation of model backdooring, and the approach is reasonable and well-explained

Weaknesses: there's a mix of concerns, including novelty and setting (a few works that dont quite overlap, but are close + realisticness of a fairness data poisoning attack), baselines (effectiveness of group targeting)

**Justification For Why Not Higher Score:**

This one ends up borderline (despite the low review scores), but ultimately I think the reviewers' confusion about the novelty and various baselines should probably be handled via a more extensive rewrite.

**Justification For Why Not Lower Score:**

N/A

---

### Decision · Program_Chairs · 2024-01-16

Reject